# A simplified pneumonia severity index (PSI) for clinical outcome prediction in COVID-19

Shu-Ching Chang[1,¤,*], Gary L. Grunkemeier[2], Jason D. Goldman[3,4,5], Mansen Wang[6], Paul A. McKelvey[7], Jennifer Hadlock[8], Qi Wei[8], George A. Diaz[9,10,11]

1 Providence St. Joseph Health, Portland, Oregon, United States of America, 2 Division of Cardiothoracic Surgery, Oregon Health & Science University, Portland, OR, United States of America, 3 Division of Infectious Diseases, Swedish Medical Center, Seattle, WA, United States of America, 4 Swedish Center for Research and Innovation, Swedish Medical Center, Seattle, WA, United States of America, 5 Division of Allergy and Infectious Diseases, University of Washington, Seattle, WA, United States of America, 6 ClinChoice, Portland, OR, United States of America, 7 Providence Heart Institute, Providence St. Joseph Health, Portland, Oregon, United States of America, 8 Institute for Systems Biology, Seattle, Washington, United States of America, 9 Division of Medicine, Section of Infectious Diseases, Providence Regional Medical Center Everett, Everett, WA, United States of America, 10 Washington State University Elson S. Floyd College of Medicine, Spokane, WA, United States of America, 11 Providence Research Network, Renton, WA, United States of America

☯ These authors contributed equally to this work.
¤ Current address: The Society of Thoracic Surgeons (STS), Chicago, IL, United States of America
* schang@sts.org

**Data Availability Statement:** All relevant data are within the manuscript and its Supporting Information files.

## Abstract

### Background

The Pneumonia Score Index (PSI) was developed to estimate the risk of dying within 30 days of presentation for community-acquired pneumonia patients and is a strong predictor of 30-day mortality after COVID-19. However, three of its required 20 variables (skilled nursing home, altered mental status and pleural effusion) are not discretely available in the electronic medical record (EMR), resulting in manual chart review for these 3 factors. The goal of this study is to compare a simplified 17-factor version (PSI-17) to the original (denoted PSI-20) in terms of prediction of 30-day mortality in COVID-19.

### Methods

In this retrospective cohort study, the hospitalized patients with confirmed SARS-CoV-2 infection between 2/28/20–5/28/20 were identified to compare the predictive performance between PSI-17 and PSI-20. Correlation was assessed between PSI-17 and PSI-20, and logistic regressions were performed for 30-day mortality. The predictive abilities were compared by discrimination, calibration, and overall performance.

### Results

Based on 1,138 COVID-19 patients, the correlation between PSI-17 and PSI-20 was 0.95. Univariate logistic regression showed that PSI-17 had performance similar to PSI-20, based on AUC, ICI and Brier Score. After adjusting for confounding variables by multivariable

**Funding:** The author(s) received no specific funding for this work.

**Competing interests:** I have read the journal's policy and the authors of this manuscript have the following competing interests: G.A.D reported receipt of clinical trial research support from Gilead Sciences, Regeneron, Roche, Boehringer Ingelheim, Edesa Biotech and NeuroBo Pharmaceuticals. J.D.G reports contracted research from Helix, Gilead, Eli Lilly, and Regeneron, grants from Merck (BARDA) and Gilead, and collaborative services agreements with Adaptive Biotechnologies, Monogram Biosciences and Labcorp; and serving as a consultant, speaker or advisory board member for Gilead, and Eli Lilly. All other authors have nothing to disclose.

logistic regression, PSI-17 and PSI-20 had AUCs (95% CI) of 0.85 (0.83–0.88) and 0.86 (0.84–0.89), respectively, indicating no significant difference in AUC at significance level of 0.05.

## Conclusion

PSI-17 and PSI-20 are equally effective predictors of 30-day mortality in terms of several performance metrics. PSI-17 can be obtained without the manual chart review, which allows for automated risk calculations within an EMR. PSI-17 can be easily obtained and may be a comparable alternative to PSI-20.

## Introduction

The Pneumonia Severity Index (PSI), developed to predict mortality in patients with bacterial pneumonia, risk stratifies which patients can safely discharge to home [1]. Recently, PSI has also been validated in viral pneumonia [2]. We and others have validated the PSI for hospitalized patients with COVID-19 pneumonia as a predictor of mortality [3,4]. While the prediction of mortality is important, the PSI score is most helpful for deciding if a patient can safely return home and has been utilized for this purpose extensively in Emergency Department settings. During the COVID-19 pandemic, hospitals in the US resorted to crisis standards of patient care [5] due to lack of hospital beds and other critical resources. Current standard of care for admitted patients with COVID-19 requiring low flow oxygen is a combination of remdesivir and dexamethasone [6,7]. Current practice for patients presenting without hypoxemia is discharge to home, possibly with enrollment into a telehealth home monitoring program [8], and provision of antiviral therapy if presenting early in disease.

The PSI score contains 20 variables; 3 of them–pleural effusion (PE), altered mental status (AMS), and skilled nursing facility (SNF)–are not easily extracted from discrete fields in the electronic medical record (EMR). In this study, we define PSI-17, the PSI score without these 3 variables, and use a data set of 1,138 COVID-19 patients from 13 hospitals in the Providence St. Joseph Health system to compare its performance to the full PSI score (PSI-20).

Since PSI-17 can be abstracted and calculated easily from the EMR, it can provide to front line caregivers the ability to identify which hypoxemic patients with COVID-19 pneumonia can safely be treated in an outpatient setting. This study aims to compare the simplified 17-factor version (PSI-17) to the original PSI-20 in terms of prediction of 30-day mortality based on 1,138 patients who were hospitalized from in the first pandemic wave with confirmed SARS-CoV-2 infection [3].

## Methods

### Study population

In this retrospective cohort study, the hospitalized patients with confirmed SARS-CoV-2 infection between 2/28/20–5/28/20 that met the same inclusion and exclusion criteria as in our previous study (Diaz et al., 2022) were used to evaluate the predictive performance between PSI-17 and PSI-20. Data was accessed for research purposes from the electronic medical records via the PSJH electronic data warehouse or by manual record review from July 12th, 2020 to June 21st, 2021, and reviewed by the study team prior to analysis. Authors had access to limited

identifiable information during data collection. Detailed information about data collection and study population can be found in Diaz et al. (2022).

## Statistical analysis

The components and scores of PSI-17 and PSI-20 are summarized in **S1 Table**. Correlation of PSI-17 and PSI-20 was assessed using Spearman's correlation coefficient. The predictive values of the PSI indices were evaluated by logistic regression models of 30-day mortality with each index as the predictor variable. The estimates of the models were given by odds ratio (OR) with 95% confidence intervals (95% CI) for each 10 points of the PSI score. We assessed the prediction performance of PSI-17 versus PSI-20 based on discrimination, calibration, and overall performance [9].

For model discrimination, areas under the receiver operating characteristic (ROC) curve (AUC) were estimated. In general, an AUC of 0.5 suggests no discrimination (i.e., ability to diagnose patients with and without the disease based on the test), 0.7 to 0.8 is considered acceptable, 0.8 to 0.9 is considered excellent, and more than 0.9 is considered outstanding [10]. Calibration plots with integrated calibration index (ICI), which was estimated by the weighted average absolute difference between observed and predicted probabilities [11] were used for the assessment of the agreement between observed outcomes and predictions. For overall performance, reflecting discrimination and calibration simultaneously, the Akaike Information Criterion (AIC), the Brier Score and the net reclassification index (NRI) [12] were also calculated, with 95% CI constructed using 1,000 bootstrap replications. All tests were two-sided and statistical significance was set at $p < 0.05$. Statistical analyses were performed using the R software, version 4.1.3 (R Core Team, 2022).

## Patient consent statement

Waiver of consent for this study was approved by the Providence St. Joseph institutional review board (STUDY2020000143).

## Results

In total, 1,138 patients who were hospitalized from 2/28/20–5/28/20 with confirmed SARS-CoV-2 infection were included in the study cohort (**S2 Table**) [3]. All 17 PSI components were extracted from the EMR. SNF, AMS and PE were collected from manual chart review by study investigators. **S1 Fig** shows the number of patients for the 3 chart-review variables and their combinations. The median (Q1 –Q3) values for PSI-17, and PSI-20 were 72 (54–90) and 77 (55–102), respectively with a correlation coefficient of 0.95. Univariate logistic regression analyses showed that PSI-17 and PSI-20 were significant predictors of 30-day mortality, with similar effect sizes (**Table 1**).

## Prediction performance of PSI-17 versus PSI-20

Predictive performance was similar between PSI-17 and PSI-20 based on different metrics and measures (**Table 1**). The OR (95% CI) was 1.51 (1.40–1.63) and 1.43 (1.35–1.52) for each 10-points of the score for PSI-17 and PSI-20, respectively. The AUC for 30-day mortality was 0.79 (0.76–0.82) for PSI-17 and was 0.81 (0.78–0.84) for PSI-20, which are acceptable and similar (**Fig 1A**), with Brier scores (scaled) for both models reflecting good overall prediction (**Table 1**). The relationship between observed and predicted probabilities for 30-day mortality in calibration plots showed good agreement with predicted probabilities over the entire empirical range of the predicted probabilities (**Fig 2A and 2B**). The 30-day mortality by PORT Class

**Table 1. Comparison of predictive performance for 30-day mortality between PSI-17 and PSI-20 using univariate, and risk-adjusted multivariable logistic regression models.**

| Performance measure | Unadjusted | P-value | Risk-adjusted[a] | P-value |
|---|---|---|---|---|
| *Model Prediction*, OR (95% CI) | | | | |
| PSI-17 (per 10 points) | 1.51 (1.40–1.63) | NS[b] | 1.40 (1.28–1.53) | NS[b] |
| PSI-20 (per 10 points) | 1.43 (1.35–1.52) | | 1.39 (1.29–1.50) | |
| *Discrimination*, AUC (95% CI) | | | | |
| PSI-17 | 0.79 (0.76–0.82) | NS[b] | 0.85 (0.83–0.88) | NS[b] |
| PSI-20 | 0.81 (0.78–0.84) | | 0.86 (0.84–0.89) | |
| *Calibration*, ICI: | | | | |
| PSI-17 | 0.027 (0.02–0.04) | NS[b] | 0.021 (0.01–0.03) | NS[b] |
| PSI-20 | 0.025 (0.02–0.04) | | 0.019 (0.01–0.03) | |
| *Overall*, Brier Score, scaled | | | | |
| PSI-17 | 0.128 (0.09–0.18) | NS[b] | 0.226 (0.19–0.31) | NS[b] |
| PSI-20 | 0.156 (0.11–0.21) | | 0.248 (0.21–0.33) | |

Abbreviations: OR, odds ratio; NS, not significant; CI, confidence interval, AUC, area under the curve, ICI, integrated calibration index.

[a]Adjusted for WHO-OSS, do-not-resuscitate status, race/ethnicity, body mass index, creatinine clearance <50 mL/min, dementia, hypertension, D-dimer level, absolute lymphocyte count, any corticosteroid use, and a term for temporal effect.

[b]NS; Not statistically significant at significance level of 0.05.

[1] further demonstrates both PSI scores as predictors of 30-day mortality with COVID-19, with no significant differences in 30-day mortality between PSI-17 versus PSI-20 within each PORT Class (Class I: 1.2% vs. 1.3%, p = 0.99; Class II: 5.7% vs. 3.4%, p = 0.20; Class III: 19.0% vs. 14.8%, p = 0.23; Class IV: 34.5% vs. 28.2%, p = 0.12; Class V: 42.1% vs. 47.1%, p = 0.89) (**Fig 3**).

After including the 11 risk factors identified by Diaz et al. (2022) [3] (WHO-OSS, do-not-resuscitate status, race/ethnicity, body mass index, creatinine clearance <50 mL/min, dementia, hypertension, D-dimer level, absolute lymphocyte count, any corticosteroid use, and a

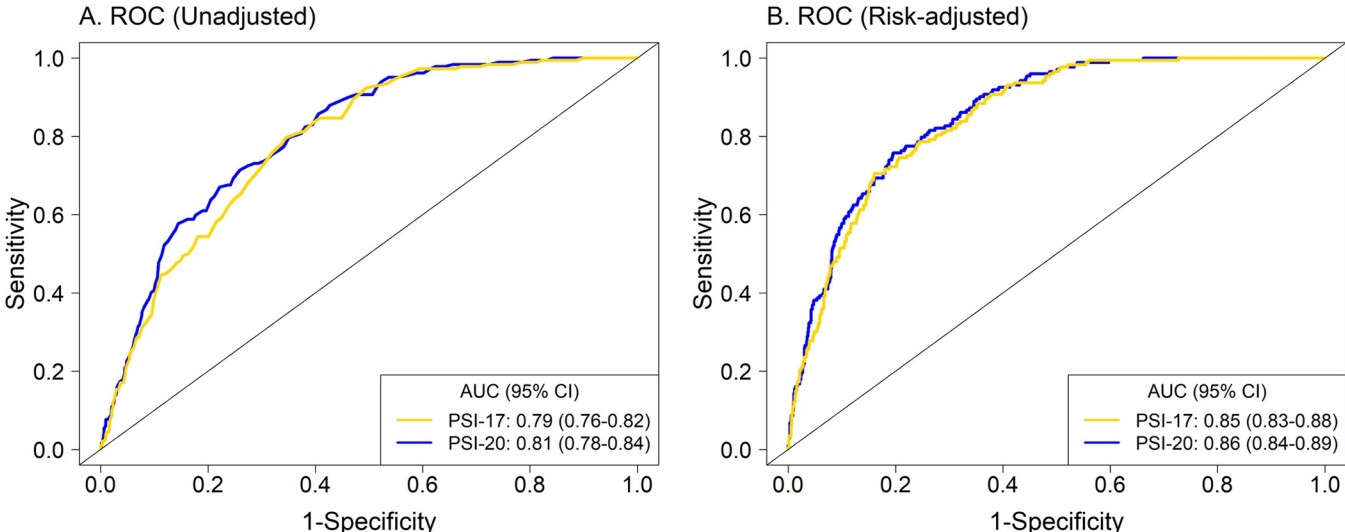

**Fig 1.** The comparison of ROCs from (A) univariate logistic regression and (B) risk-adjusted multivariable logistic regression, adjusted for 11 risk factors for 30-day mortality.

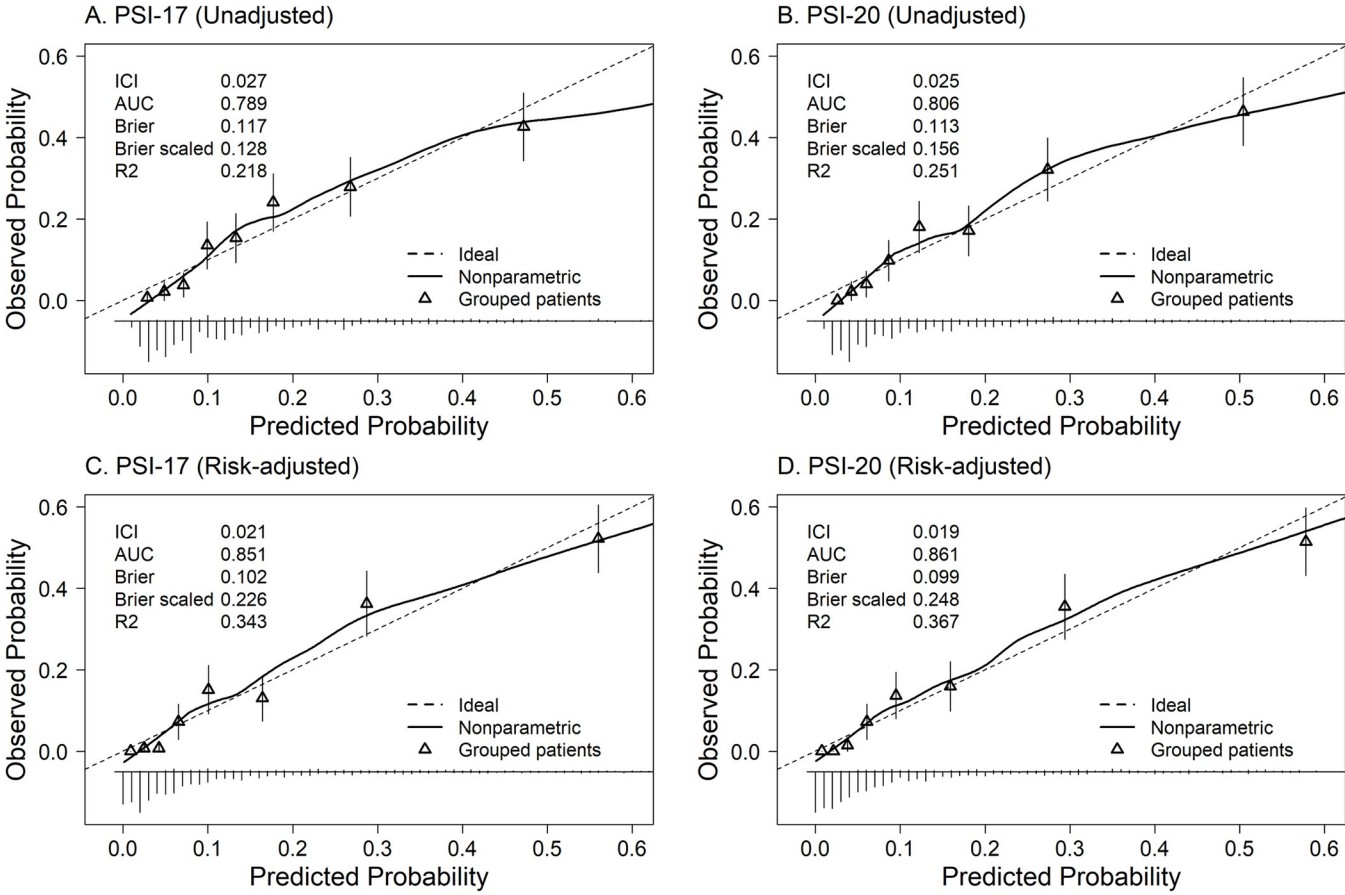

**Fig 2.** Calibration plots of mean observed rates versus predicted rates of 30-day mortality for univariate logistic regression on (A and B) and risk-adjusted multivariable logistic regression adjusted for 11 risk factors (C and D). Samples were equally divided into 8 groups, according to their predicted 30-day mortality probability. "Rug plot" below the main figures are histograms showing the number of observations for the corresponding predicted 30-day mortality probability.

term for temporal effect), the AUCs (95% CIs) of the models increased to 0.85 (0.83–0.88) for PSI-17 and 0.86 (0.84–0.89) for PSI-20 (**Fig 1B**). In addition, the calibration plots showed that both indices had excellent agreement between observed and predicted 30-day mortality probabilities (**Fig 2C and 2D**).

The net reclassification index (NRI), **Fig 4,** gives an individual determination per patient for which score performed better, given the actual outcome [12]. The net proportion of patients who didn't die within 30-days and were assigned a lower risk by PSI-20 was 0.5, while the net proportion of patients who died within 30-days and were assigned a higher risk by the PSI-20 risk model was -0.044. Thus, although PSI-20 had a 50% improvement in the predictions of survivors, it introduced a few more errors in the predictions of 30-day death.

## Discussion

Community-Acquired Pneumonia (CAP) is an acute lung infection that causes 1.5 million hospitalizations in the United States each year [13]. A Recent study showed that the hospitalized COVID-19 patients with diabetes have an increased risk for pneumonia, intensive care unit requirement, intubation, and death [14]. As the COVID pandemic continues to evolve, health systems could see subsequent waves with large numbers of patients admitted to the

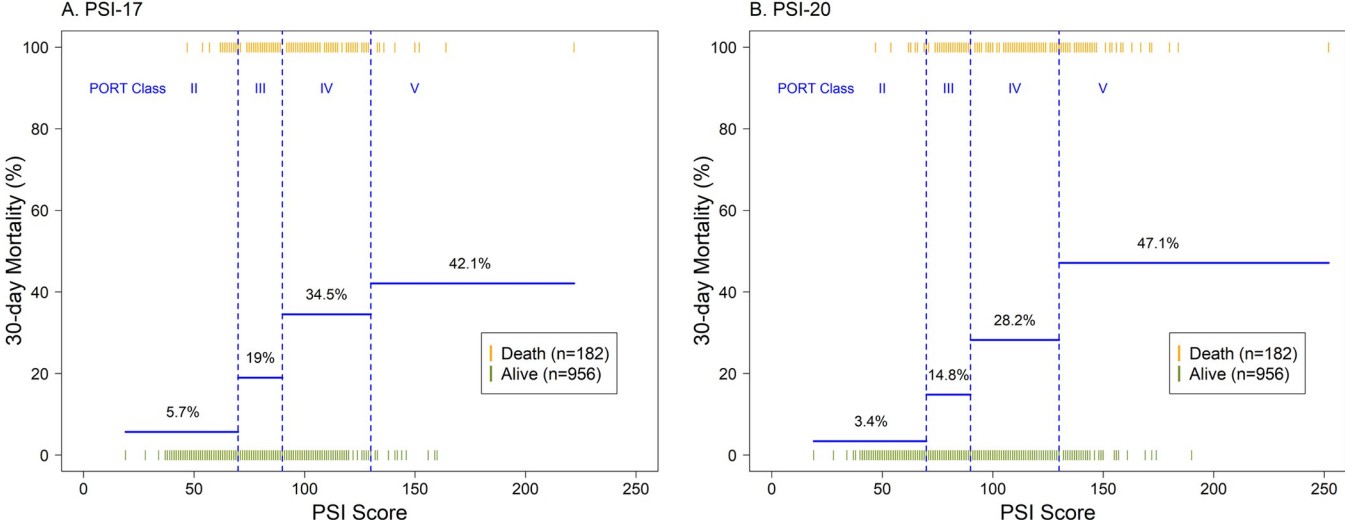

**Fig 3. 30-day mortality by PORT Class defined by Fine et al. (1997): PORT class II: ≤70; PORT class III: 71–90; PORT class IV: 91–130; PORT class V: >130.** 30-day mortality rates for PORT Class I are 1.2% and 1.3% for PSI-17 and PSI-20, respectively. Individual patient PSI scores who survived or died are shown in green or yellow, respectively.

inpatient setting with COVID-19 pneumonia, consuming valuable resources. There are currently a variety of therapeutics available for patients with mild to moderate COVID-19 [15]. Once patients develop new hypoxemia, common practice is hospital admission for treatment with intravenous remdesivir and dexamethasone [16]. We have previously described that PSI-20 is a good predictor of mortality from COVID-19 pneumonia [3]. Using PSI-20 can assist physicians to determine if outpatient treatment is appropriate for community acquired pneumonia, based on predicted mortality [1]. However, using PSI-20 can be cumbersome as it requires manual data entry for 3 fields and therefore is used variably in clinical practice. In this study, we demonstrate that the PSI-17 can accurately predict mortality in COVID-19. PSI-17 includes data elements which are found discreetly in the EMR and is amenable to automatic calculation within the EMR.

It is likely that COVID-19 patients with low risk of mortality who are also hypoxemic may be safely managed in the outpatient setting, according to current treatment recommendations, particularly in combination with telehealth home monitoring and provision of antivirals. Absolute mortality differed in the PORT classes defined by Fine et al. (1997) for bacterial pneumonia, compared to this study for COVID-19 pneumonia. For instance, for bacterial pneumonia in Fine et al. (1997), PORT class II predicted 0.6–0.9% mortality where outpatient care is reasonable barring other factors affecting care and PORT class III predicts a 0.9–2.8% mortality where outpatient or inpatient care is reasonable, depending on clinical judgement. In this study, PORT class I predicted 1.2% and 1.3% mortality and PORT class II predicted 5.7% and 3.4% mortality, from PSI-17 and PSI-20, respectively. Given the high mortality seen in the first wave of the pandemic, triage recommendations could be shifted for PORT classes. For instance, PORT class I would be reasonable to triage to outpatient care, and PORT class II would be reasonable for outpatient or inpatient care, depending on clinical judgement. To determine whether PSI-17 is predictive of mortality in community acquired pneumonia [13], validation in an appropriate cohort would need to be performed.

Similarly, as the mortality has decreased over time through evolving SARS-CoV-2 variants [17–19], the application of a mortality prediction tool would need to be validated in different variant eras. Future study could determine different recommendations for use as mortality

## Components of Net Reclassification Index

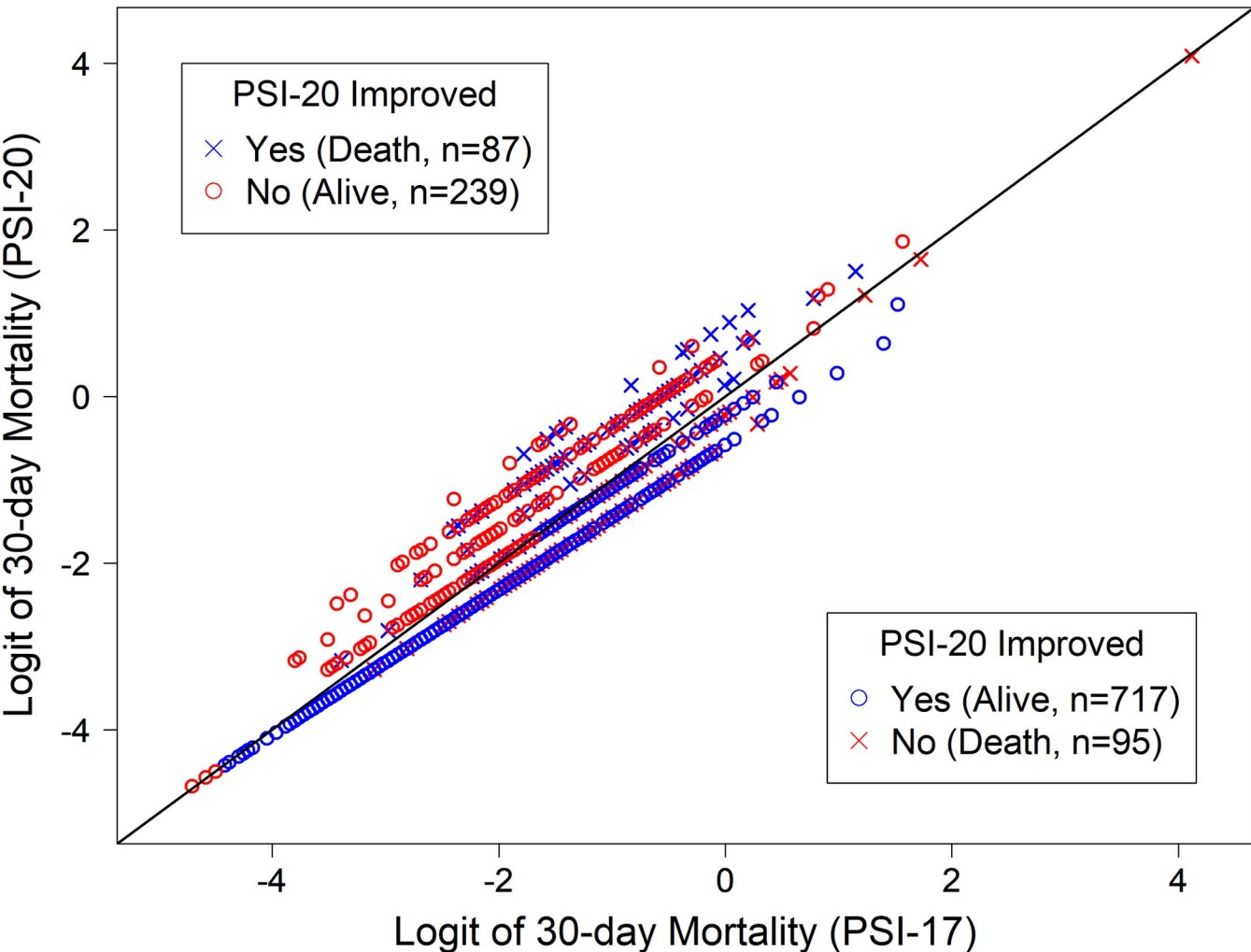

**Fig 4. Scatter plot of the components needed for the net reclassification index.** Open circles and crosses indicate alive and death, respectively. Blue symbols mean that PSI-20 improved PSI-17 since it had a higher predicted probability of 30-day mortality for those who died, or a lower predicted probability for those who were alive. Red symbols mean that PSI-20 didn't improve PSI-17 since it had a lower predicted probability of 30-day mortality for those who died, or a higher predicted probability for those who were alive.

during the pandemic continues to evolve given changing virulence of variants and population level protection from vaccination and prior infection.

Additionally, PSI-17 may also potentially predict if a patient is likely to decline and become hypoxemic if they are initially on room air on presentation. Based on our COVID-19 patient cohort, we showed that PSI-17 is not only efficient to calculate but also remains as an effective predictor for 30-day mortality. Jones et al. [20] took this even further and decided that if the computer was going to create PSI-17 (they call this ePSI) then the computer might as well add more terms for 30-day mortality prediction for pneumonia patients. In this paper, we also augmented the PSI-17 with other clinical variables for 30-day mortality. After adjusting for the 11 risk factors, we identified previously [3], the AUC's of the PSI-17 and PSI-20 models both increased to above 0.85. The PSI is not comprehensive and other model features could be added, such as diabetes, which is a known risk factor for poor outcomes in COVID-19 [14]

and is not one of the listed co-morbidities in PSI (S1 Table). Pneumonia predictive scores have been studied in COVID-19 and PSI was shown as one of the better performing scores for 30-day COVID mortality along with CURB-65, and covid specific scores: 4C and COVID GRAM among 11 different scores for mortality assessment [21].

In this study, we have shown that PSI-17 may be an easily obtained and effective alternative to PSI-20 for mortality prediction in COVID-19 as determined in this early pandemic cohort. The simplification of PSI-17 to remove fields that require manual chart abstraction can serve for integration of scores into clinical interfaces of electronic health records to be accessed at the point of care, or for controlling of confounders in retrospective studies of COVID-19. However, use of PSI-17 may not be relevant in the individual assessment of a COVID-19 patient where these three variables are readily obtainable. External validation as well as updated assessment of clinical utility in subsequent calendar years (including in the era of additional therapies) merits future study to further evaluate the association of automated calculation of PSI-17, disposition, and outcomes.

## Supporting information

**S1 Fig. Proportional-area Venn diagram.** Among 1,138 patients, 433 (38%) patients had at least one of the 3 PSI-20 variables (SNF, skilled nursing facility; AMS, altered mental status; PE, pleural effusion); 705 patients (62%) did not have any of the 3 variables.
(TIF)

**S1 Table. Points assigned for PSI-20 (Fine et al., 1997) and PSI-17.**
(DOCX)

**S2 Table. Patient demographics and baseline characteristics.**
(DOCX)

**S1 Data.**
(XLSX)

## Acknowledgments

We acknowledge the patients included in this study and the caregiver teams across the Providence organization. SCC was affiliated with the Providence St. Joseph Health at the time of this study and is currently affiliated with the Society of Thoracic Surgeons (STS).

## Author Contributions

**Conceptualization:** Shu-Ching Chang, Gary L. Grunkemeier, Jason D. Goldman, Mansen Wang, Paul A. McKelvey, Jennifer Hadlock, Qi Wei, George A. Diaz.

**Data curation:** Shu-Ching Chang, Paul A. McKelvey.

**Formal analysis:** Shu-Ching Chang.

**Funding acquisition:** George A. Diaz.

**Investigation:** Shu-Ching Chang, Gary L. Grunkemeier, Jason D. Goldman, George A. Diaz.

**Methodology:** Shu-Ching Chang, Gary L. Grunkemeier, Jason D. Goldman, Mansen Wang, George A. Diaz.

**Writing – original draft:** Shu-Ching Chang, George A. Diaz.

**Writing – review & editing:** Shu-Ching Chang, Gary L. Grunkemeier, Jason D. Goldman, Mansen Wang, Paul A. McKelvey, Jennifer Hadlock, Qi Wei, George A. Diaz.

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
