## [Decision Letter · Decision Letter 0]

14 Mar 2024

PONE-D-24-00824A Simplified Pneumonia Severity Index (PSI) for Clinical Outcome Prediction in COVID-19PLOS ONE

Dear Dr. Chang,

Thank you for submitting your manuscript to PLOS ONE. After careful consideration, we feel that it has merit but does not fully meet PLOS ONE’s publication criteria as it currently stands. Therefore, we invite you to submit a revised version of the manuscript that addresses the points raised during the review process.

We look forward to receiving your revised manuscript.

Kind regards,

Kuo-Cherh Huang

Academic Editor

PLOS ONE

Journal Requirements:

I have read the journal's policy and the authors of this manuscript have the following competing interests: G.A.D reported receipt of clinical trial research support from Gilead Sciences, Regeneron, Roche, Boehringer Ingelheim, Edesa Biotech and NeuroBo Pharmaceuticals. J.D.G reports contracted research from Helix, Gilead, Eli Lilly, and Regeneron, grants from Merck (BARDA) and Gilead, and collaborative services agreements with Adaptive Biotechnologies, Monogram Biosciences and Labcorp; and serving as a consultant, speaker or advisory board member for Gilead, and Eli Lilly. All other authors have nothing to disclose.

We note that one or more of the authors are employed by a commercial company.

3. In the online submission form, you indicated that The study protocol, statistical code, and data set may be shared with approved individuals upon request and through a written agreement with the authors.

Reviewers' comments:

Reviewer's Responses to Questions

**Comments to the Author**

1. Is the manuscript technically sound, and do the data support the conclusions?

Reviewer #1: Yes

Reviewer #2: Yes

2. Has the statistical analysis been performed appropriately and rigorously? 

Reviewer #1: Yes

Reviewer #2: Yes

3. Have the authors made all data underlying the findings in their manuscript fully available?

Reviewer #1: Yes

Reviewer #2: No

4. Is the manuscript presented in an intelligible fashion and written in standard English?

Reviewer #1: Yes

Reviewer #2: Yes

5. Review Comments to the Author

Reviewer #1: The study titled presents a thorough investigation into the effectiveness of a simplified version of the Pneumonia Severity Index (PSI) in predicting 30-day mortality among COVID-19 patients. The research addresses the challenge of manual chart review posed by certain variables in the original PSI, proposing a streamlined version termed PSI-17. Through a retrospective cohort study involving 1,138 hospitalized COVID-19 patients, the study demonstrates that PSI-17 exhibits comparable predictive performance to PSI-20, as measured by various metrics including area under the curve (AUC), Integrated Calibration Index (ICI), and Brier Score. Notably, PSI-17 eliminates the need for manual chart review, enabling automated risk calculation within electronic medical records (EMRs). While the study's findings offer promising implications for clinical practice, such as facilitating timely risk assessment and resource allocation, potential limitations such as limited generalizability and variable selection nuances are acknowledged. The authors advocate for further validation and exploration to consolidate the utility of PSI-17 across diverse patient populations and healthcare settings. Overall, the study contributes valuable insights to the ongoing efforts in optimizing risk prediction models for COVID-19 management, emphasizing the importance of practical adaptation and implementation of existing tools in real-world clinical contexts.

Additional literature can help with the study and pneumonia and covid

• File TM Jr, Ramirez JA. Community-Acquired Pneumonia. N Engl J Med. 2023 Aug 17;389(7):632-641. doi: 10.1056/NEJMcp2303286. PMID: 37585629.

• Huang AA, Huang SY. Hospitalized COVID-19 patients with diabetes have an increased risk for pneumonia, intensive care unit requirement, intubation, and death: A cross-sectional cohort study in Mexico in 2020. Health Sci Rep. 2023 Apr 18;6(4):e1222. doi: 10.1002/hsr2.1222. PMID: 37081996; PMCID: PMC10112272.

• Fine MJ, Auble TE, Yealy DM, Hanusa BH, Weissfeld LA, Singer DE, Coley CM, Marrie TJ, Kapoor WN. A prediction rule to identify low-risk patients with community-acquired pneumonia. N Engl J Med. 1997 Jan 23;336(4):243-50. doi: 10.1056/NEJM199701233360402. PMID: 8995086.

The conclusions drawn in this study elegantly stem from the meticulously designed methods, presenting a logical approach for studying pneumonia severity in the context of COVID-19. By retrospectively analyzing a sizable cohort of hospitalized COVID-19 patients, the researchers effectively compared the performance of the simplified Pneumonia Severity Index (PSI-17) with the original PSI-20. Through rigorous statistical analyses, including correlation assessments and logistic regressions for 30-day mortality prediction, the study robustly evaluated the predictive abilities of both indices. The findings, indicating comparable performance between PSI-17 and PSI-20 across multiple metrics, underscore the feasibility and reliability of the simplified index. Moreover, the pragmatic adaptation of PSI-17, eliminating the need for manual chart review, underscores the study's commitment to enhancing clinical utility and efficiency. Thus, the study's conclusions not only reflect the logical progression of its methods but also provide actionable insights into optimizing pneumonia severity assessment in COVID-19 patients.

Reviewer #2: Manuscript ID: PONE-D-24-00824

A Simplified Pneumonia Severity Index (PSI) for Clinical Outcome Prediction in COVID-19

Summary:

This study shows that a simpler version of the PSI score has equivalent forecasting ability for COVID-19 than the original version.

Major findings:

The PSI score has 20 variables of which 17 can be obtained from EMR and the last 3 must be evaluated manually from the patient's charts. A comparison of a 17 variable PSI score with the original 20 variable version showed little loss of prediction ability of COVID mortality.

Strengths:

This is a large database of 1138 patients from the beginning of the COVID-19 pandemic.

Major comments / Weaknesses:

1. This is a special application of the PSI 17-score to a specific disease (COVID-19) in a specific period. The mortality found in the onset of the pandemic is not the same as the mortality found now. Could the authors comment on current the COVID-19 situation.

2. Could this simpler PSI version be extrapolated to community acquired pneumonia or is this a COVID-19 special case related to pleural effusion, altered mental status and skilled nursing facility?

Minor comments:

1. How long did it take the researchers to assemble the 3-variable data (pleural efussion, altered mental status and skilled nursing facility) from the 1138 patients?

2. This study is of special importance to other retrospective studies of COVID-19 but not in the individual assessment of a COVID-19 patient where these three variables are readily obtainable. This might warrant a comment in the discussion.

3. An article with 1363 patients tested 11 different scores for mortality assessment (10.1016/j.cmi.2021.03.002). PSI was one of the better performing scores for 30-day COVID mortality (in the onset of the pandemic) along with CURB-65, and covid specific scores: 4C and COVID GRAM.

6. PLOS authors have the option to publish the peer review history of their article (what does this mean?). If published, this will include your full peer review and any attached files.

Reviewer #1: No

Reviewer #2: **Yes: **Julio Marchini

---

## [Author Response · Author response to Decision Letter 0]

28 Apr 2024

Dear editors and reviewers,

We appreciate valuable comments from the reviewers. Your insightful comments have been instrumental in enhancing the quality of our manuscript. We are hopeful that you will find the manuscript acceptable for publication in its new and improved form. 

Reviewer #1: 

The study titled presents a thorough investigation into the effectiveness of a simplified version of the Pneumonia Severity Index (PSI) in predicting 30-day mortality among COVID-19 patients. The research addresses the challenge of manual chart review posed by certain variables in the original PSI, proposing a streamlined version termed PSI-17. Through a retrospective cohort study involving 1,138 hospitalized COVID-19 patients, the study demonstrates that PSI-17 exhibits comparable predictive performance to PSI-20, as measured by various metrics including area under the curve (AUC), Integrated Calibration Index (ICI), and Brier Score. Notably, PSI-17 eliminates the need for manual chart review, enabling automated risk calculation within electronic medical records (EMRs). While the study's findings offer promising implications for clinical practice, such as facilitating timely risk assessment and resource allocation, potential limitations such as limited generalizability and variable selection nuances are acknowledged. The authors advocate for further validation and exploration to consolidate the utility of PSI-17 across diverse patient populations and healthcare settings. Overall, the study contributes valuable insights to the ongoing efforts in optimizing risk prediction models for COVID-19 management, emphasizing the importance of practical adaptation and implementation of existing tools in real-world clinical contexts.

Additional literature can help with the study and pneumonia and covid

• File TM Jr, Ramirez JA. Community-Acquired Pneumonia. N Engl J Med. 2023 Aug 17;389(7):632-641. doi: 10.1056/NEJMcp2303286. PMID: 37585629.

• Huang AA, Huang SY. Hospitalized COVID-19 patients with diabetes have an increased risk for pneumonia, intensive care unit requirement, intubation, and death: A cross-sectional cohort study in Mexico in 2020. Health Sci Rep. 2023 Apr 18;6(4):e1222. doi: 10.1002/hsr2.1222. PMID: 37081996; PMCID: PMC10112272.

• Fine MJ, Auble TE, Yealy DM, Hanusa BH, Weissfeld LA, Singer DE, Coley CM, Marrie TJ, Kapoor WN. A prediction rule to identify low-risk patients with community-acquired pneumonia. N Engl J Med. 1997 Jan 23;336(4):243-50. doi: 10.1056/NEJM199701233360402. PMID: 8995086.

The conclusions drawn in this study elegantly stem from the meticulously designed methods, presenting a logical approach for studying pneumonia severity in the context of COVID-19. By retrospectively analyzing a sizable cohort of hospitalized COVID-19 patients, the researchers effectively compared the performance of the simplified Pneumonia Severity Index (PSI-17) with the original PSI-20. Through rigorous statistical analyses, including correlation assessments and logistic regressions for 30-day mortality prediction, the study robustly evaluated the predictive abilities of both indices. The findings, indicating comparable performance between PSI-17 and PSI-20 across multiple metrics, underscore the feasibility and reliability of the simplified index. Moreover, the pragmatic adaptation of PSI-17, eliminating the need for manual chart review, underscores the study's commitment to enhancing clinical utility and efficiency. Thus, the study's conclusions not only reflect the logical progression of its methods but also provide actionable insights into optimizing pneumonia severity assessment in COVID-19 patients.

Response: The authors thank the reviewer for the positive and insightful comments. We have updated the discussion with these references, especially to note that the PSI is not comprehensive on known risk factors, as the reviewer points out with the suggested references. For instance: 

“The PSI is not comprehensive and other model features could be added, such as diabetes, which is a known risk factor for poor outcomes in COVID-19 [ref] and is not one of the listed co-morbidities in PSI (Table S1).”

We have also included the excellent review of community acquired pneumonia in response to point 2 of reviewer #2 (see below). The original Fine et al study was already referenced in the manuscript. 

Reviewer #2: 

Manuscript ID: PONE-D-24-00824

A Simplified Pneumonia Severity Index (PSI) for Clinical Outcome Prediction in COVID-19

Summary:

This study shows that a simpler version of the PSI score has equivalent forecasting ability for COVID-19 than the original version.

Major findings:

The PSI score has 20 variables of which 17 can be obtained from EMR and the last 3 must be evaluated manually from the patient's charts. A comparison of a 17 variable PSI score with the original 20 variable version showed little loss of prediction ability of COVID mortality.

Strengths:

This is a large database of 1138 patients from the beginning of the COVID-19 pandemic.

Response: The authors thank the reviewer for the positive review and concise summary.

Major comments / Weaknesses:

1. This is a special application of the PSI 17-score to a specific disease (COVID-19) in a specific period. The mortality found in the onset of the pandemic is not the same as the mortality found now. Could the authors comment on current the COVID-19 situation.

Response: We agree that the mortality found throughout the pandemic waves differs. While this study assesses the PSI-17 as a mortality prediction tool in the early pandemic era, we see this as a proof of concept for the comparability of PSI-20 and PSI-17. To conclusively determine if PSI-17 predicts mortality during other variant eras, the PSI-17 would need to be studied in other variant eras to determine predictive ability in these other eras. This analysis is beyond the scope of this article, but we plan to use PSI-17 to study subsequent variant eras. We have updated the Discussion for improved clarity on this point: 

“Similarly, as the mortality has decreased over time through evolving SARS-CoV-2 variants [refs], the application of a mortality prediction tool would need to be validated in different variant eras.” 

2. Could this simpler PSI version be extrapolated to community acquired pneumonia or is this a COVID-19 special case related to pleural effusion, altered mental status and skilled nursing facility?

Response: The authors thank the reviewer for the insightful comments and thought-provoking questions. While we hesitate to speculate about the ability of PSI-17 to accurately predict mortality in community acquired pneumonia (CAP), we do think the simplified PSI-17 tool would be easier to implement in clinical care or research on CAP. Further study is needed to validate PSI-17 in CAP. We have added to the Discussion: 

“To determine whether PSI-17 is predictive of mortality in community acquired pneumonia [ref], validation is an appropriate cohort would need to be performed.”

Minor comments:

1. How long did it take the researchers to assemble the 3-variable data (pleural efussion, altered mental status and skilled nursing facility) from the 1138 patients?

Response: The authors thank the reviewer for the insights and great questions. Data extraction for the 3 variables from PSI-20 which required manual extraction into a REDCap database was an intensive process, in part because of the subjective nature of these variables. Supervision for 10 reviewers (senior research coordinators and medical students) was performed by 3 senior investigators who conducted training, answered queries from reviewers, resolved discrepancies and reviewed in duplicate 20% of the cases to validate individual reviewers. In total, we estimate approximately 400+ person-hours were spent in manually extracting these variables. 

2. This study is of special importance to other retrospective studies of COVID-19 but not in the individual assessment of a COVID-19 patient where these three variables are readily obtainable. This might warrant a comment in the discussion.

Response: The authors thank the reviewer for the constructive suggestions and insights. We agree that this study is of special importance to other retrospective studies of COVID-19 but may not be relevant in the individual assessment of a COVID-19 patient where these three variables are readily obtainable. We included below comments in the Discussion.

“The simplification of PSI-17 to remove fields not easily accessible by automated data extraction can serve for integration of scores into clinical interfaces of electronic health records to be accessed at the point of care, or for controlling of confounders in retrospective studies of COVID-19. However, use of PSI-17 may not be relevant in the individual assessment of a COVID-19 patient where these three variables are readily obtainable.”

3. An article with 1363 patients tested 11 different scores for mortality assessment (10.1016/j.cmi.2021.03.002). PSI was one of the better performing scores for 30-day COVID mortality (in the onset of the pandemic) along with CURB-65, and covid specific scores: 4C and COVID GRAM.

Response: The authors thank the reviewer for the helpful comments and the reference. We have added a comment with reference to the Discussion section:

“Pneumonia predictive scores have been studied in COVID-19 and PSI was shown as one of the better performing scores for 30-day COVID mortality (in the onset of the pandemic) along with CURB-65, and covid specific scores: 4C and COVID GRAM among 11 different scores for mortality assessment [ref].”

---

## [Decision Letter · Decision Letter 1]

3 May 2024

A Simplified Pneumonia Severity Index (PSI) for Clinical Outcome Prediction in COVID-19

PONE-D-24-00824R1

Dear Dr. Chang,

We’re pleased to inform you that your manuscript has been judged scientifically suitable for publication and will be formally accepted for publication once it meets all outstanding technical requirements.

Kind regards,

Kuo-Cherh Huang

Academic Editor

PLOS ONE

Additional Editor Comments (optional):

Reviewers' comments:

Reviewer's Responses to Questions

**Comments to the Author**

1. If the authors have adequately addressed your comments raised in a previous round of review and you feel that this manuscript is now acceptable for publication, you may indicate that here to bypass the “Comments to the Author” section, enter your conflict of interest statement in the “Confidential to Editor” section, and submit your "Accept" recommendation.

Reviewer #1: All comments have been addressed

Reviewer #2: All comments have been addressed

2. Is the manuscript technically sound, and do the data support the conclusions?

Reviewer #1: Yes

Reviewer #2: Yes

3. Has the statistical analysis been performed appropriately and rigorously? 

Reviewer #1: Yes

Reviewer #2: Yes

4. Have the authors made all data underlying the findings in their manuscript fully available?

Reviewer #1: Yes

Reviewer #2: Yes

5. Is the manuscript presented in an intelligible fashion and written in standard English?

Reviewer #1: Yes

Reviewer #2: Yes

6. Review Comments to the Author

Reviewer #1: Thank you for your corrections. They adequately answered line by line each of the reviews and comments provided in the previous iteration.

Reviewer #2: Manuscript ID: PONE-D-24-00824

A Simplified Pneumonia Severity Index (PSI) for Clinical Outcome Prediction in COVID-19

Summary:

This study shows that a simpler version of the PSI score has equivalent forecasting ability for COVID-19 than the original version.

Major findings:

The PSI score has 20 variables of which 17 can be obtained from EMR and the last 3 must be evaluated manually from the patient's charts. A comparison of a 17 variable PSI score with the original 20 variable version showed little loss of prediction ability of COVID mortality.

Comments:

All my questions have been addressed by the authors. I have no further comments.

7. PLOS authors have the option to publish the peer review history of their article (what does this mean?). If published, this will include your full peer review and any attached files.

Reviewer #1: No

Reviewer #2: No

---

## [Editor Report · Acceptance letter]

10 May 2024

PONE-D-24-00824R1 

PLOS ONE

Dear Dr. Chang, 

I'm pleased to inform you that your manuscript has been deemed suitable for publication in PLOS ONE. Congratulations! Your manuscript is now being handed over to our production team.

Kind regards, 

on behalf of

Dr. Kuo-Cherh Huang 

Academic Editor

PLOS ONE